# Longitudinal assessment of COVID-19 vaccine immunogenicity in people with HIV stratified by CD4+ T-cell count in the Netherlands: A two-year follow-up study

Marlou J. Jongkees[1], Susanne Bogers[2], Rory D. de Vries[2], Corine H. GeurtsvanKessel[2], Pedro Miranda Afonso[3,4], Kathryn S. Hensley[1], Bart J. A. Rijnders[1], Kees Brinkman[5], Casper Rokx[1☉], Anna H. E. Roukens[6☉*]

1 Department of Internal Medicine, Section Infectious Diseases, and Department of Medical Microbiology and Infectious Diseases, Erasmus University Medical Centre, Rotterdam, The Netherlands, 2 Department of Viroscience, Erasmus University Medical Centre, Rotterdam, The Netherlands, 3 Department of Biostatistics, Erasmus University Medical Centre, Rotterdam, The Netherlands, 4 Department of Epidemiology, Erasmus University Medical Centre, Rotterdam, The Netherlands, 5 Department of Internal Medicine and Infectious Diseases, OLVG Hospital, Amsterdam, The Netherlands, 6 Department of Infectious Diseases, Leiden University Medical Centre, Leiden, The Netherlands

☉ These authors contributed equally to this work.
* A.H.E.Roukens@lumc.nl

## Abstract

### Background

Although guidelines for COVID-19 additional vaccination strategies generally prioritise people with advanced HIV infection, recommendations vary globally, with some countries recommending an annual vaccination for all people with HIV (PWH), while others restrict this to PWH with a CD4+ T-cell count < 200 cells per µL.

### Methods

We conducted a prospective cohort study in 448 adult PWH. The primary outcome was the SARS-CoV-2 spike (S1)-specific IgG antibody level at 1, 6, 12, 18, and 24 months after completing a primary COVID-19 vaccination series (two doses of BNT162b2, mRNA-1273, or ChAdOx1-S, or one dose of Ad26.COV2.S). We compared the antibody kinetics over two years between PWH with a baseline CD4+ T-cell count < 200 cells per µL (n = 16) vs. ≥ 200 cells per µL (n = 432) with a mixed-effects model. Secondary outcomes included variables associated with the kinetics of S1-specific antibody levels and the incidence of breakthrough infections.

### Results

The median most recent CD4+ T-cell count prior to primary vaccination was 140 (IQR 80–165) in the < 200 cells per µL group, and 688 (IQR 520–899) in the ≥ 200 cells per µL group at the time of primary vaccination. S1-specific antibodies were lower in PWH

**Data availability statement:** Deidentified individual participant data that underlie the results reported in this article, along with the analytic code, and other supporting documents, will be made available to researchers who submit a methodologically sound proposal. Data requests may be sent to the Erasmus MC HIV Eradication Group (EHEG) at eheg@erasmusmc.nl.

**Funding:** This study was funded by the Netherlands Organisation for Health Research and Development (ZonMw – https://www.zonmw.nl) [grant number 10430072010008]. The grant was awarded to the institutions involved, rather than to individual researchers. The funder had no role in study design, data collection and analysis, decision to publish, or preparation of the manuscript.

**Competing interests:** I have read the journal's policy and the authors of this manuscript have the following competing interests: RDdV is supported by the Health~Holland grant EMCLHS20017, co-funded by the PPP Allowance made available by the Health~Holland, Top Sector Life Sciences & Health, to stimulate public-private partnerships, and is listed as an inventor of the fusion inhibitory lipopeptide [SARSHRCPEG4]2-chol in a provisional patent application. KSH has received support for attending meetings and travel from Gilead. BJAR declares the receipt of research grants from Gilead and MSD and honoraria for advisory boards from AstraZeneca, Roche, Gilead, and F2G. KB has received research and educational grants from ViiV and Gilead, as well as consulting fees for advisory boards from ViiV, Gilead, MSD, and AstraZeneca. CR has received research grants from ViiV, Gilead, ZonMW, AIDSfonds, Erasmus MC, and Health~Holland, and honoraria for advisory boards from Gilead and ViiV. AR has received grants from the Bill and Melinda Gates Foundation and the Leids Universitair Fonds, participated on the board of an investor-initiated clinical trial on convalescent plasma for COVID-19, and is the chief editor of the Dutch Journal of Infectious Diseases and a member of the European Medicines Agency expert group on vaccines. This does not alter our adherence to PLOS ONE policies on sharing data and materials. The other authors have declared that no competing interests exist.

with a CD4+ T-cell count < 200 vs. ≥ 200 cells per µL during the two-year follow-up, with predicted S1-specific antibody levels of 514 (95% CI 456–578) vs. 2758 (95% CI 1488–5110) BAU per mL at 12 months (p < 0.001) and 839 (95% CI 732–959) vs. 3505 (95% CI 1712–7175) BAU per mL at 24 months (p < 0.001). The overall incidence of SARS-CoV-2 infections was 55% and comparable between groups. A CD4+ T-cell count < 200 cells per µL, higher age, and a vector-based primary vaccination series were negatively associated with S1-specific antibody levels over time.

## Conclusion

Long-term humoral responses were lower in PWH with a CD4+ T-cell count < 200 cells per µL compared to those with a CD4+ T-cell count ≥ 200 cells per µL. National COVID-19 vaccine guidelines recommending booster vaccines for all PWH, should therefore specifically emphasise the need for booster vaccines in those with a CD4+ T-cell count < 200 cells per µL.

**Trial registration:** The trial was registered on the International Clinical Trials Platform (registration number: EUCTR2021-001054-57-N).

## Introduction

Since the beginning of the pandemic, higher coronavirus disease 2019 (COVID-19) mortality has been observed in immunocompromised individuals, including people with HIV (PWH) (RR 1.20, 95% CI 1.05–1.36) [1]. Although disease severity has decreased with the emergence of the Omicron variant, immunocompromised individuals remain at an increased risk of severe COVID-19 [2]. During the Omicron era, a higher risk of hospitalisation due to COVID-19 has been reported in PWH compared to HIV-negative individuals [3].

Short-term immune responses following a primary COVID-19 vaccination series were lower in PWH compared to HIV-negative individuals [4–6], particularly in those with a CD4+ T-cell count < 200 cells per µL. Larger studies indicate that this association between CD4+ T-cell count and antibody response persisted after booster vaccination [7–9]. An additional mRNA-1273 vaccination was highly effective in seroconverting PWH with a non-response after the primary vaccination series [10], and both monovalent and bivalent booster vaccinations improved neutralising antibody responses in PWH [11–13].

Currently, COVID-19 vaccination studies in PWH do not exceed one year of follow-up [14–16]. In addition to the lack of data on the durability of the immune response, risk factors for accelerated waning in PWH have not been assessed. For other vaccines, such as yellow fever and hepatitis B, a shorter duration of protection has been reported in PWH compared to HIV-negative individuals, with a lower CD4+ T-cell count associated with reduced levels of neutralising antibodies [17–19].

Understanding long-term immune responses is important as the need for yearly COVID-19 booster vaccines for at-risk groups is being discussed worldwide. The lack

of long-term data is reflected in the inconsistency of national guidelines regarding COVID-19 booster vaccination recommendations for PWH. While the Centers for Disease Control and Prevention in the US recommend a twice-yearly booster vaccination for PWH with a CD4+ T-cell count < 200 cells per µL [20], European guidelines generally recommend annual vaccination for all PWH, irrespective of their CD4+ T-cell count [21,22]. We hypothesise that PWH with a low CD4+ T-cell count exhibit lower spike (S)-specific antibody levels over time compared to PWH with a normal CD4+ T-cell count. Therefore, this study aimed to evaluate the durability of S-specific antibodies over a two-year period following a primary COVID-19 vaccination series among PWH.

## Methods

### Study design and participants

We performed a two-year follow-up of our nationwide prospective observational COVID-19 vaccination study among PWH in the Netherlands (COVIH) [6]. In this follow-up study, participants from the three COVIH coordinating HIV treatment centres (Erasmus Medical Centre, Leiden University Medical Centre, and OLVG Hospital) were monitored longitudinally for two years after completing the primary COVID-19 vaccination series. The inclusion criteria were a minimum age of 18 years and a confirmed HIV infection. PWH were stratified into two subgroups based on their most recent CD4+ T-cell count (CD4+ T-cell count < 200 vs. ≥ 200 cells per µL) prior to the primary vaccination series.

### Clinical procedures

Between 22 February and 7 September 2021, 432 participants were included in the three HIV treatment centres. Following the primary vaccination series, participants were monitored through routine care for two years. Since 98% of participants had a CD4+ T-cell count ≥ 200 cells per µL and 79% had a CD4+ T-cell count > 500 cells per µL, we actively sought to include PWH with a CD4+ T-cell count < 500 cells per µL, who had not previously been enrolled in the study. Participants with a CD4+ T-cell count < 500 were identified via treating HIV physicians in one of the HIV treatment centres (Erasmus Medical Centre). To minimise selection bias, no information regarding prior COVID-19 vaccinations, severe acute respiratory syndrome coronavirus 2 (SARS-CoV-2) infection history, or spike protein S1 subunit (S1)-specific IgG antibodies were available at the time of inclusion. In total, 16 additional participants with a CD4+ T-cell count < 500 cells per µL were included retrospectively between 13 October 2023 and 11 April 2024, including eight participants with a CD4+ T-cell count < 200 cells per µL. Between January and October 2021, participants received the primary vaccination series in accordance with manufacturers' regulations as part of the Dutch COVID-19 vaccination campaign. The primary vaccination series consisted of two doses of BNT162b2, mRNA-1273, or ChAdOx1-S, or one dose of Ad26.COV2.S. All booster vaccinations administered were mRNA-based, including mRNA-1273, BNT162b2, mRNA-1273.214, BNT162b2 Omicron BA.1, mRNA-1273.222, BNT162b2 Omicron BA.5, mRNA-1273.815, and BNT162b2 Omicron XBB.1.5. The type of booster vaccination was determined by availability in the Dutch COVID-19 vaccination campaign and was independent of the primary vaccination series. S1-specific IgG antibodies were measured in blood samples taken during twice-yearly HIV check-ups. Clinical data were collected in an electronic case record file and included the following: year of birth, sex assigned at birth, current use of combination antiretroviral therapy (cART), most recent plasma HIV-RNA load (copies per mL), most recent CD4+ T-cell count (cells per µL) at the time of the primary vaccination series, CD4+ T-cell counts measured during the subsequent two years (cells per µL), nadir CD4+ T-cell count (cells per µL), use of immunosuppressive therapy, diagnosis of solid organ neoplasia or haematological malignancy within 12 months prior to the primary vaccination series, prior COVID-19 vaccinations, and history of self-reported breakthrough SARS-CoV-2 infections confirmed by positive PCR or rapid antigen tests. Dates and types of prior COVID-19 vaccinations, as well as dates of intercurrent SARS-CoV-2 infections, including the test type, whether PCR or rapid antigen test, and infection severity, were obtained through yearly online questionnaires and telephone surveys.

## Laboratory procedures

The concentrations of IgG binding antibodies specific to the ancestral S1-subunit were measured using a validated quantitative chemiluminescence immunoassay, the LIAISON SARS-CoV-2 TrimericS IgG assay (DiaSorin, Saluggia, Italy), with a lower limit of quantification of 4.81 binding antibody units (BAU) per mL, at the Erasmus University Medical Centre, a WHO SARS-CoV-2 reference laboratory.

## Outcomes

The primary outcome was the geometric mean of S1-specific IgG antibodies in PWH at 1, 6, 12, 18, and 24 months after primary vaccination. The kinetics of S1-specific IgG antibodies over two years were compared between PWH with a CD4 + T-cell count < 200 vs. ≥ 200 cells per µL. Secondary outcomes included HIV-related and HIV-unrelated variables associated with the kinetics of S1-specific IgG antibodies, as well as the incidence of self-reported SARS-CoV-2 breakthrough infections, confirmed by a positive PCR or rapid antigen test.

## Statistical analysis plan

The power calculation for the primary endpoint of the COVIH study was reported separately [6] and no power calculation was performed for this predefined secondary endpoint. All participants from the three COVIH coordinating centres in the original trial were eligible to participate in the current study. Data were described using number (percentage) or median (interquartile range [IQR]). A linear mixed-effects model was used to compare the kinetics of S1-specific IgG antibodies over the two-year period between PWH with a most recent CD4+ T-cell count < 200 vs. ≥ 200 cells per µL at the time of the primary vaccination series. The association between HIV-related and HIV-unrelated variables and the kinetics of S1-specific IgG antibodies was analysed using a second model that included the following fixed effects: quadratic effect of time since the primary vaccination series, sex, age (as a continuous quantitative variable), most recent CD4+ T-cell count at baseline (< 200 vs. ≥ 200 cells per µL), nadir CD4+ T-cell count since HIV diagnosis, type of COVID-19 primary vaccination series (mRNA vs. vector), S1-specific IgG antibody levels measured 4–6 weeks after the primary vaccination series, and time since the last COVID-19 (booster) vaccination dose or last SARS-CoV-2 infection. Additionally, the kinetics of S1-specific IgG antibodies over time were compared between PWH with a nadir CD4+ T-cell count < 200 vs. ≥ 200 cells per µL. A likelihood ratio test was used to compare the longitudinal evolution of S1-specific IgG antibodies between PWH with a most recent CD4+ T-cell count < 200 and those with ≥ 200 cells per µL, as well as between PWH with a nadir CD4+ T-cell count < 200 vs. ≥ 200 cells per µL. Comparison of the SARS-CoV-2-infection-free probability between the groups was performed using a log-rank test. The proportion of PWH with a second or third infection was compared between both groups using a Chi-squared test. For all analyses, S1-specific IgG antibodies were log10-transformed. P values < 0.05 were considered statistically significant. Undetectable S1-specific IgG antibody responses (< 4.81 BAU per mL) were reported as 4.81 in the statistical analyses. Data were analysed using RStudio 4.3.2 and the nlme package (v. 3.1.162).

## Ethics statement

The trial was conducted in accordance with the Declaration of Helsinki, Good Clinical Practice guidelines, and the Dutch Medical Research Involving Human Subjects Act (WMO). It was approved by the Medical Ethics Committees United Nieuwegein (MEC-U, reference 20.125) and registered on the International Clinical Trials Platform (EUCTR2021-001054-57-N). All participants provided written informed consent. This study adhered to the Strengthening the Reporting of Observational Studies in Epidemiology (STROBE) guidelines to ensure comprehensive reporting of the data.

## Results

### Baseline characteristics

Of the 448 participants enrolled in the COVIH follow-up study, 16 participants had a most recent CD4$^+$ T-cell count < 200 cells per µL at the time of the primary vaccination series (3.6%) (Fig 1). Over the two-year follow-up period, nine of the 16 participants in the baseline CD4 < 200 group showed an increase to ≥ 200 cells per µL, while three of the 432 participants in the baseline CD4 ≥ 200 group experienced a decrease to < 200 cells per µL. Ten participants dropped out during follow-up: four died and six moved to another hospital. A total of 2142 samples were analysed, including 77 samples from participants in the baseline CD4 < 200 group and 2065 from participants in the baseline CD4 ≥ 200 group. Overall, the questionnaire response rate was 90.0%.

The cohort was predominantly male (87.1%), with a median age of 56 years (IQR 47–63) (Table 1). Participants in the baseline CD4 < 200 group had a median most recent CD4$^+$ T-cell count of 140 cells per µL (IQR 80–165) and a median nadir CD4$^+$ T-cell count of 25 cells per µL (IQR 10–125). In contrast, participants in the baseline CD4 ≥ 200 group had a median most recent CD4$^+$ T-cell count of 688 cells per µL (IQR 520–899) and a median nadir CD4$^+$ cell count of 230 cells per µL (IQR 103–330). All participants were on combination antiretroviral therapy (cART), except for one elite controller. Almost all participants were virally suppressed, with 98.2% having a plasma HIV viral load < 50 copies per mL at inclusion. Most participants had received BNT162b2 (77.5%) or ChAdOx1-S (15.0%) for their primary vaccination series, while a smaller proportion had received mRNA-1273 (6.2%) or Ad26.COV2.S (1.3%).

### Kinetics of antibody responses

S1-specific IgG antibody levels were lower in PWH with a baseline CD4$^+$ T-cell count < 200 cells per µL compared to those with a baseline CD4$^+$ T-cell count ≥ 200 cells per µL in the two years following the primary vaccination series (p < 0.001) (Fig 2). Predicted S1-specific IgG antibodies levels were lower in the baseline CD4 < 200 group compared to the baseline CD4 ≥ 200 group at the following time points: 168 (95% CI 151–187) vs. 1504 (95% CI 858–2636) BAU per mL at month 6 (p < 0.001), 514 (95% CI 456–578) vs. 2758 (95% CI 1488–5110) BAU per mL at month 12 (p < 0.001), 879 (95% CI 784–987) vs. 3660 (95% CI 2008–6671) BAU per mL at month 18 (p < 0.001) and 839 (95% CI 732–959) vs. 3505 (95% CI 1712–7175) BAU per mL at month 24 (p < 0.001), respectively. S1-specific IgG antibody levels of the 16 participants who were retrospectively included are indicated in S1 Fig. The median number of booster vaccine doses received after completing the primary vaccination series was 2 (IQR 1–3) in both groups, with the type of COVID-19 booster vaccines BNT162b2 vs. mRNA-1273 being comparable between the groups.

### Variables associated with the kinetics of antibody responses

A CD4$^+$ T-cell count < 200 cells per µL (p < 0.001), older age (p = 0.001), a vector-based primary vaccination series (p < 0.001), and increased time since the last COVID-19 vaccination dose or SARS-CoV-2 infection (p < 0.001) were associated with lower S1-specific IgG antibody levels over time (S2 Table). Higher S1-specific IgG antibody levels one month after the primary vaccination series (p < 0.001) were positively associated with S1-specific IgG antibody levels over time. No association was found for sex or nadir CD4$^+$ T-cell count. Additionally, a nadir CD4$^+$ T-cell count < 200 vs. ≥ 200 cells per µL was not independently associated with S1-specific IgG antibody levels over time (S2 Fig).

### Incidence of SARS-CoV-2 infection

No significant difference was observed in the probability of remaining free of the first SARS-CoV-2 infection between PWH with a CD4$^+$ T-cell count < 200 cells per µL (46.7% (95%CI 27.2–80.2)) and those with a CD4$^+$ T-cell count ≥ 200

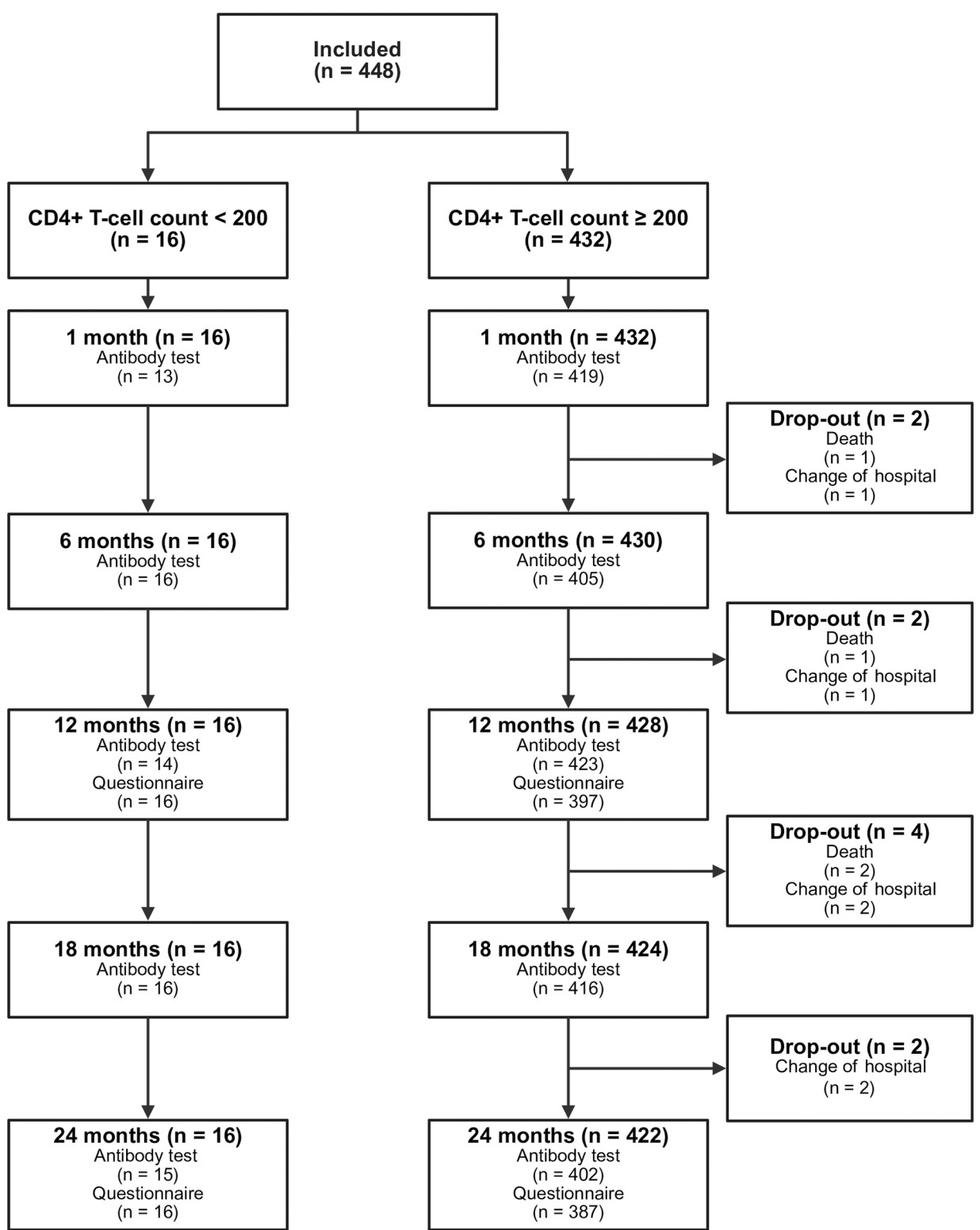

**Fig 1. Flowchart of included PWH in the two-year follow-up of the COVIH study.** *n* denotes the number of unique individuals.

**Table 1. Baseline characteristics of participants stratified by CD4+ T-cell count.**

| | All PWH N = 448 | PWH with a CD4+ T-cell count < 200 cells per µL N = 16 | PWH with a CD4+ T-cell count ≥ 200 cells per µL N = 432 |
|---|---|---|---|
| **Age**, years[a] | 56 (47–63) | 55 (45–59) | 56 (47–63) |
| **Male** | 390 (87.1%) | 13 (81.3%) | 377 (87.3%) |
| **Female** | 58 (12.9%) | 3 (18.7%) | 55 (12.7%) |
| **On cART** | 447 (99.8%) | 16 (100%) | 431 (99.8%) |
| **Use of immunosuppressive therapy**[b] | 7 (1.6%) | 1 (6.2%) | 6 (1.4%) |
| **Malignancy in the past 12 months**[a] | 2 (0.4%) | 0 | 2 (0.5%) |
| **Most recent plasma HIV viral load < 50**, copies per mL | 440 (98.2%) | 14 (87.5%) | 426 (98.6%) |
| **Most recent plasma HIV viral load < 200**, copies per mL | 446 (99.6%) | 15 (93.8%) | 431 (99.8%) |
| **Most recent CD4+ T-cell count,** cells per µL | 675 (509–894) | 140 (80–165) | 688 (520–899) |
| **Nadir CD4+ T-cell count**, cells per µL | 220 (84–320) | 25 (10–125) | 230 (103–330) |
| **COVID-19 vaccine type for primary vaccination series** | | | |
| BNT162b2 | 347 (77.5%) | 13 (81.3%) | 334 (77.3%) |
| mRNA-1273 | 28 (6.2%) | 2 (12.5%) | 26 (6.0%) |
| ChAdOx1-S | 67 (15.0%) | 1 (6.2%) | 66 (15.3%) |
| Ad26.COV2.S | 6 (1.3%) | 0 | 6 (1.4%) |
| **COVID-19 vaccinations received**[c] | | | |
| Primary series only | 31 (7.7%) | 2 (13.3%) | 29 (7.5%) |
| Primary series + 1 booster | 103 (25.6%) | 4 (26.7%) | 99 (25.5%) |
| Primary series + 2 boosters | 139 (34.5%) | 4 (26.7%) | 136 (35.1%) |
| Primary series + 3 boosters | 121 (30.0%) | 5 (33.3%) | 115 (29.6%) |
| Primary series + 4 boosters | 9 (2.2%) | 0 | 9 (2.3%) |
| **History of at least one SARS-CoV-2 infection**[c,d] | | | |
| Yes | 225 (54.9%) | 10 (62.5%) | 215 (54.6%) |
| No | 185 (45.1%) | 6 (37.5%) | 179 (45.4%) |
| **Dominant variant in the Netherlands at the time of first reported SARS-CoV-2 infection** [23] | | | |
| Alpha variant[c] | 7 (3.1%) | 1 (10.0%) | 6 (2.8%) |
| Delta variant[c] | 24 (10.7%) | 0 | 24 (11.2%) |
| Omicron variant[c] | 194 (86.2%) | 9 (90.0%) | 185 (86.0%) |
| **History of a second SARS-CoV-2 infection**[c,d] | 37 (9.0%) | 2 (12.5%) | 35 (8.9%) |
| **History of a third SARS-CoV-2 infection**[c,d] | 5 (1.2%) | 1 (6.3%) | 4 (1.0%) |
| **S1-specific IgG antibodies 4–6 weeks after primary vaccination**, BAU per mL | 1108 (967–1270) | 65 (188–236) | 1210 (1065–1374) |

Data are n (%), median (IQR), or geometric mean (95% CI). Abbreviations: BAU, binding antibody units; cART, combination antiretroviral therapy; HIV, human immunodeficiency virus; IQR, interquartile range; S, spike.

[a]At baseline (defined as 1 January 2021).

[b]Types of immunosuppressive therapy included prednisolone, methotrexate, and mycophenolate mofetil.

[c]Missing data were excluded from denominators: COVID-19 vaccinations received: 45 (10.0%) overall, 1 (6.3%) in the < 200 group, 44 (10.2%) in the ≥ 200 group; History of SARS-CoV-2 infection: 38 (8.5%) overall, none in the baseline CD4 < 200 group, 38 (8.8%) in the baseline CD4 ≥ 200 group.

[d]Confirmed by a self-reported positive polymerase chain reaction test or rapid antigen test.

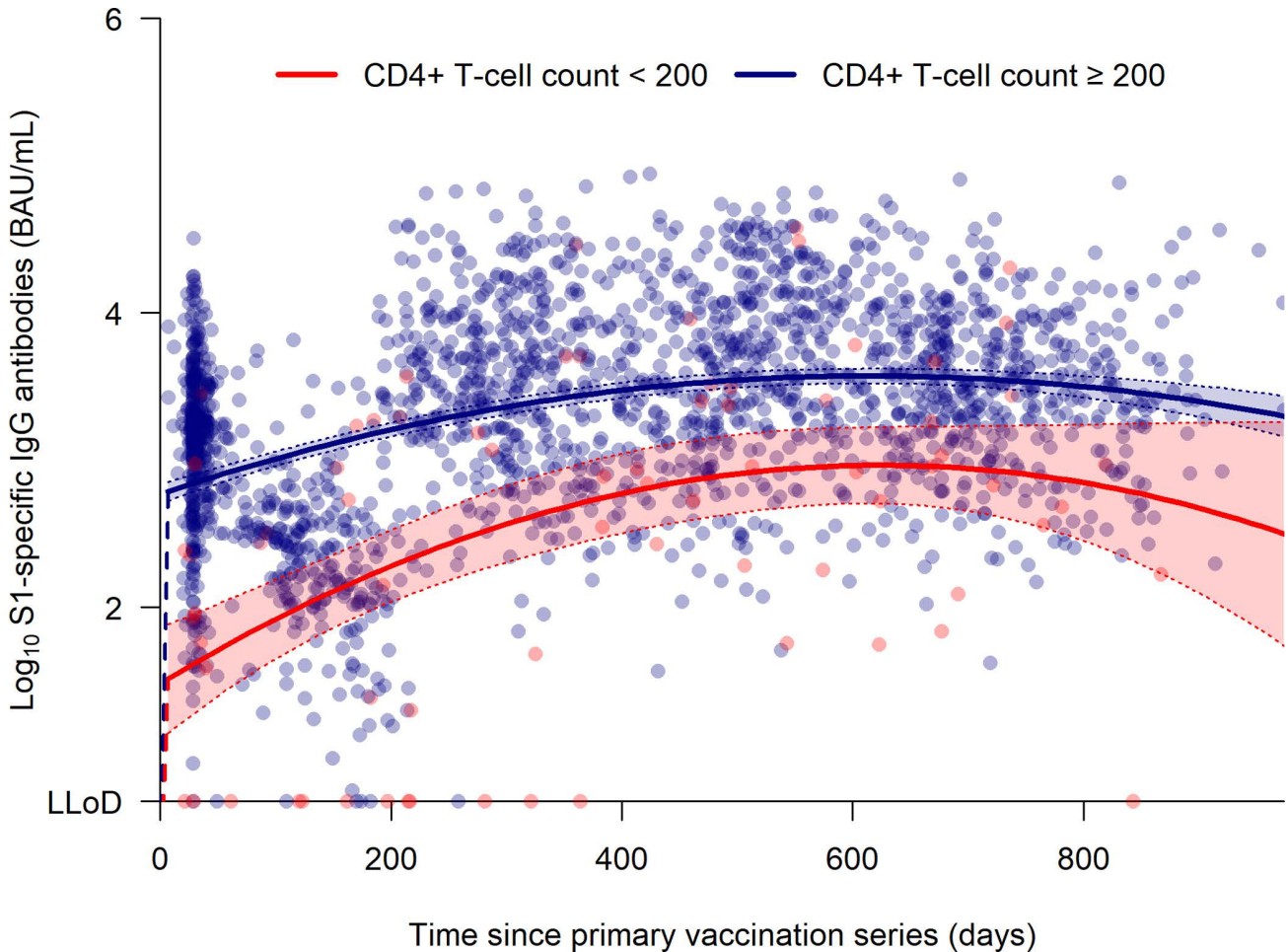

**Fig 2. SARS-CoV-2 spike (S1)-specific IgG antibody concentration (BAU per mL) in PWH stratified by CD4+ T-cell count over time (days) from immediately prior to the primary vaccination series (defined as day 0).** Comparison between PWH with a CD4+ T-cell count < 200 cells per µL (n = 16; red) and PWH with a CD4+ T-cell count ≥ 200 cells per µL (n = 432; blue) was performed using the likelihood ratio test. The solid lines represent the mixed-effects regression of the log₁₀ S1-specific IgG antibody levels per baseline CD4 group over time, with the dotted lines representing the 95% confidence interval. Abbreviations: BAU, binding antibody unit; LLoD, lower limit of detection; S, spike.

cells per µL (45.0% (95%CI 40.4–50.2)) during the two-year follow-up (p = 0.83). Fig 3 shows the corresponding cumulative risk functions. A second or third SARS-CoV-2 infection was reported in 12.5% and 6.3% of participants in the baseline CD4 < 200 group compared to 8.9% and 1.0% in the baseline CD4 ≥ 200 group (p = 0.89 and p = 0.44, respectively).

## SARS-CoV-2 infection severity

Four participants (0.9%) required treatment for a SARS-CoV-2 infection (S3 Table). One participant, a 61-year-old male with a most recent CD4+ T-cell count < 350 cells per µL, was admitted to the intensive care unit. Two other participants were hospitalised due to a SARS-CoV-2 infection: an 80-year-old male with a most recent CD4+ T-cell count ≥ 500 cells per µL, and a 64-year-old male with a most recent CD4+ T-cell count < 500 cells per µL. Lastly, a 75-year-old male with a most recent CD4+ T-cell count ≥ 500 cells per µL received nirmatrelvir/ritonavir treatment.

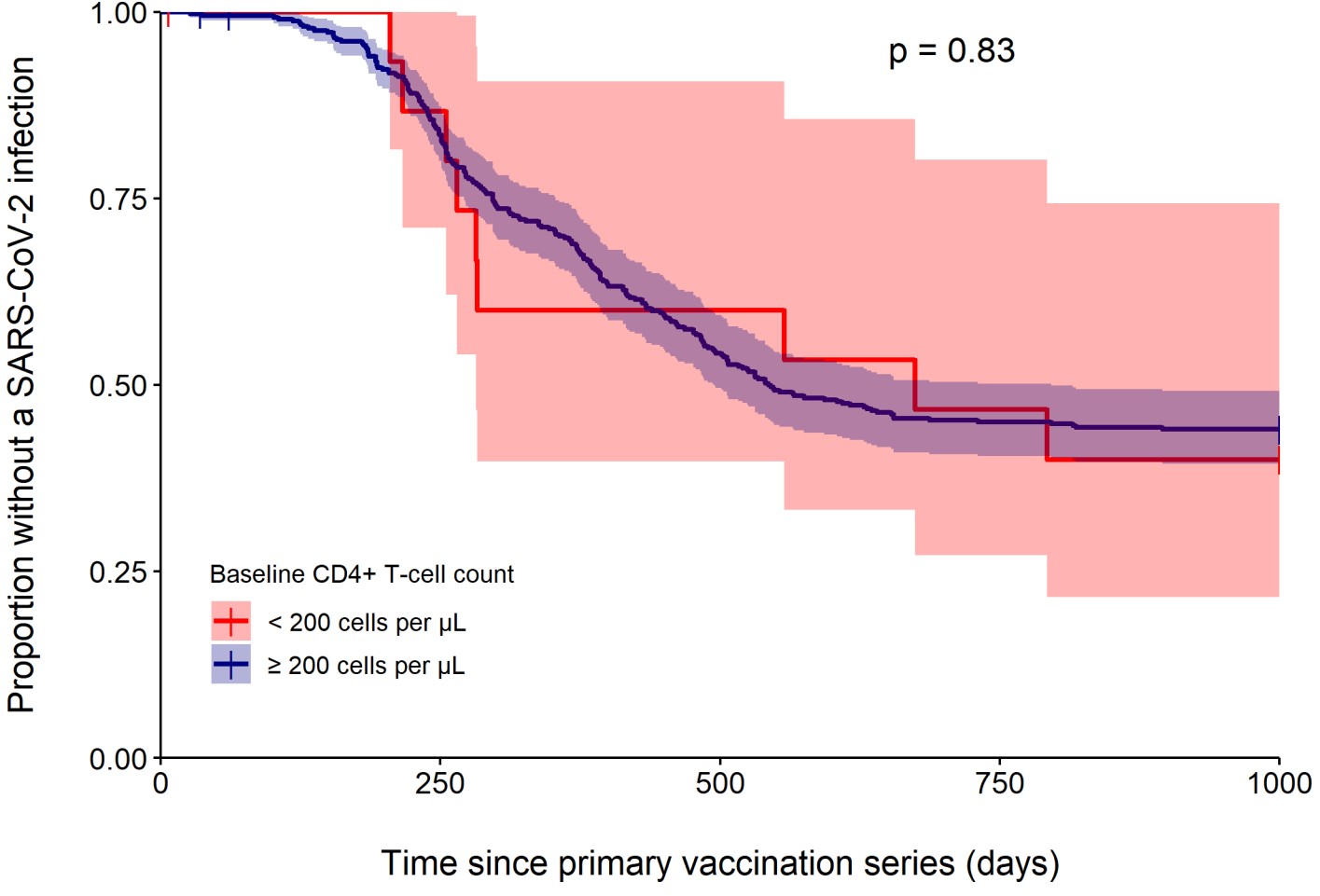

**Fig 3. Probability of remaining free of the first SARS-CoV-2 infection during the two-year follow-up in PWH stratified by CD4 ⁺ T-cell count.**
Comparison between PWH with a CD4⁺ T-cell count < 200 (n = 16; red) and PWH with a CD4⁺ T-cell count ≥ 200 (n = 432; blue) was performed by a log-rank test. Data on SARS-CoV-2 infection history was missing from 38 (8.5%) participants, of whom none in the baseline CD4 < 200 group and 38 (8.8%) in the baseline CD4 ≥ 200 group.

## Discussion

Our two-year follow-up shows that PWH with a CD4⁺ T-cell count < 200 cells per μL had consistently lower antibody levels after a primary COVID-19 vaccination series compared to PWH with a CD4⁺ T-cell count ≥ 200 cells per μL. The comparable SARS-CoV-2 infection rates and received booster doses between the groups, suggest that the difference in S1-specific antibodies is not explained by these factors.

   Despite the comparatively lower antibody levels in PWH with a CD4⁺ T-cell count < 200 cells per μL, from month six onwards, S1-specific antibodies in this group remained above 300 BAU per mL – a threshold previously correlated with the presence of neutralising antibodies against the Delta variant [24]. As the currently circulating variants from the different Omicron sublineages are antigenically even more distinct [25], breakthrough infections are expected, and were indeed observed in our cohort. An association between higher predicted spike-specific antibody levels and a lower risk of Omicron infection in both PWH and HIV-negative individuals has been demonstrated [26]. Since COVID-19 caused by SARS-CoV-2 from the Omicron sublineages is less severe, either by viral mutations or blunting by pre-existing immunity, we propose to continue the strategy of administering an annual updated booster vaccine to PWH with a CD4⁺ T-cell

count < 200 cells per μL, particularly in the case of discontinuing the yearly booster vaccine for all PWH. Given the potential for new variants to emerge, it remains important to reassess immune responses and vaccination strategies periodically, especially in individuals with advanced HIV infection.

To our knowledge, this is the first longitudinal study in PWH extending beyond one year of follow-up after COVID-19 vaccination. A key strength of our study is the large cohort size and particularly the inclusion of a sufficient number of PWH with a CD4+ T-cell count < 200 cells per μL. Another strength is the comprehensive follow-up starting from before the first COVID-19 vaccine dose, with twice-yearly blood sampling and yearly documentation of booster vaccine doses as well as SARS-CoV-2 breakthrough infections.

Our study has some limitations. In particular, we did not assess cellular immune responses or functional antibody responses, although S1-specific antibodies are known to correlate well with neutralising activity [24,27]. Additionally, almost all participants were on combination antiretroviral therapy, which limits the generalisability of our findings to regions with limited access to HIV suppressive therapy. The majority of participants in our study were male, although the sex distribution was representative of the Dutch population of PWH and comparable between the two groups. While previous studies have shown comparable humoral responses in well-treated PWH with normal CD4+ T-cell counts and HIV-negative individuals up to one year of follow-up [14–16], the absence of a control group without HIV in this study should be noted. To increase the number of participants with a low CD4+ T-cell count, 16 participants with a CD4+ T-cell count < 500 were retrospectively included prior to the first analysis. Although all eligible PWH in one of the HIV treatment centres were invited to participate and antibody levels, previous COVID-19 vaccinations, and SARS-CoV-2 infection history were unknown at the time of inclusion, retrospective inclusion could introduce selection bias. Lastly, the incidence of self-reported SARS-CoV-2 infections and COVID-19 booster vaccinations were collected through online questionnaires and telephone surveys, introducing the potential for recall bias. However, data on COVID-19 vaccinations were registered by the Dutch National Institute for Public Health and the Environment, minimising recall bias in vaccination history.

In conclusion, PWH with a CD4+ T-cell count < 200 cells per μL had lower humoral responses over the two years following a primary COVID-19 vaccination series. National COVID-19 vaccine guidelines recommending booster vaccines for all PWH should therefore specifically emphasise the need for booster vaccines in those with a CD4+ T-cell count < 200 cells per μL.

## Supporting information

**S1 Table. Overview of national COVID-19 vaccination recommendations for PWH.**
(DOCX)

**S2 Table. Linear mixed-effects model of variables associated with S1-specific IgG antibody levels over time.** The association between participant-related and vaccine-related variables on the magnitude of spike (S1)-specific IgG antibodies overall, and how the association of these variables with S1-specific IgG antibodies changes over linear and quadratic time, was evaluated by a linear mixed-effects model.
(DOCX)

**S3 Table. Characteristics of the four participants who received treatment for a SARS-CoV-2 infection.**
(DOCX)

**S1 Fig. SARS-CoV-2 spike (S1)-specific IgG antibody concentration over time (days) in 16 PWH that were included retrospectively to increase the number of participants with a CD4 + T-cell count < 500.** The 16 retrospectively included PWH are denoted by triangles (n = 8 with a CD4+ T-cell count < 200 cells per μL; dark red triangles, and n = 8 with a CD4+ T-cell count ≥ 200 cells per μL; dark blue triangles). Comparison between PWH with a CD4+ T-cell count < 200

cells per µL (n = 16; red) and PWH with a CD4+ T-cell count ≥ 200 cells per µL (n = 432; blue) was performed using the likelihood ratio test. The solid lines represent the mixed-effects regression of the $\log_{10}$ S1-specific IgG antibody levels per baseline CD4 group over time, with the dotted lines representing the 95% confidence interval. Abbreviations: BAU, binding antibody unit; LLoD, lower limit of detection; S, spike.
(TIF)

**S2 Fig. SARS-CoV-2 spike (S1)-specific IgG antibody concentration (BAU per mL) in PWH stratified by nadir CD4+ T-cell count over time (days) from immediately prior to primary vaccination (defined as day 0).** Comparison between PWH with a nadir CD4+ T-cell count < 200 cells per µL (n = 163; orange) and PWH with a nadir CD4+ T-cell count ≥ 200 cells per µL (n = 210; purple) was performed using the likelihood ratio test. The solid lines represent the mixed-effects regression of the $\log_{10}$ S1-specific IgG antibody levels per nadir CD4 group over time, with the dotted lines representing the 95% confidence interval. Missing data: nadir CD4 + T-cell count was unknown for 75 PWH. Abbreviations: BAU, binding antibody unit; LLoD, lower limit of detection; S, spike.
(TIF)

## Acknowledgments

Foremost, we thank all participants of the COVIH study. We also thank the HIV Monitoring Foundation (Stichting HIV Monitoring) for providing participant samples from the Leiden University Medical Centre. Additionally, we thank the Dutch Association of HIV-treating Physicians (NVHB) for their financial support of the poster presentation at the 25th International AIDS Conference (AIDS 2024) in Munich. Finally, we thank Aarti Soenessardien and Nieven Zhu for their valuable assistance in data collection.

## Author contributions

**Conceptualization:** Marlou J Jongkees, Rory D de Vries, Bart JA Rijnders, Kees Brinkman, Casper Rokx, Anna Helena Roukens.

**Data curation:** Marlou J Jongkees, Pedro Miranda Afonso, Kathryn S Hensley.

**Formal analysis:** Marlou J Jongkees, Pedro Miranda Afonso.

**Funding acquisition:** Bart JA Rijnders, Kees Brinkman, Casper Rokx, Anna Helena Roukens.

**Investigation:** Marlou J Jongkees, Susanne Bogers, Rory D de Vries, Corine H GeurtsvanKessel, Pedro Miranda Afonso, Kathryn S Hensley, Bart JA Rijnders, Kees Brinkman, Casper Rokx, Anna Helena Roukens.

**Methodology:** Marlou J Jongkees, Pedro Miranda Afonso, Bart JA Rijnders, Kees Brinkman, Casper Rokx, Anna Helena Roukens.

**Project administration:** Marlou J Jongkees.

**Resources:** Susanne Bogers, Rory D de Vries, Corine H GeurtsvanKessel.

**Supervision:** Bart JA Rijnders, Kees Brinkman, Casper Rokx, Anna Helena Roukens.

**Validation:** Marlou J Jongkees, Pedro Miranda Afonso.

**Visualization:** Marlou J Jongkees, Pedro Miranda Afonso.

**Writing – original draft:** Marlou J Jongkees, Casper Rokx, Anna Helena Roukens.

**Writing – review & editing:** Marlou J Jongkees, Susanne Bogers, Rory D de Vries, Corine H GeurtsvanKessel, Pedro Miranda Afonso, Kathryn S Hensley, Bart JA Rijnders, Kees Brinkman, Casper Rokx, Anna Helena Roukens.

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
