## [Decision Letter · Decision Letter 0]

21 Dec 2024

PONE-D-24-47904Longitudinal assessment of COVID-19 vaccine immunogenicity in people with HIV stratified by CD4+ T-cell count in the Netherlands: a two-year follow-up studyPLOS ONE

Dear Dr. Roukens,

Thank you for submitting your manuscript to PLOS ONE. After careful consideration, we feel that it has merit but does not fully meet PLOS ONE’s publication criteria as it currently stands. Therefore, we invite you to submit a revised version of the manuscript that addresses the points raised during the review process.

We look forward to receiving your revised manuscript.

Kind regards,

Maria Mazzitelli

Academic Editor

PLOS ONE

Journal Requirements:

This trial was funded by the Dutch Organization for Health Research and Development (ZonMw) [grant number 10430072010008].  

RDdV is supported by the Health~Holland grant EMCLHS20017, co-funded by the PPP Allowance made available by the Health~Holland, Top Sector Life Sciences & Health, to stimulate public–private partnerships, and is listed as an inventor of the fusion inhibitory lipopeptide [SARSHRC-PEG4]2-chol in a provisional patent application. KSH has received support for attending meetings and travel from Gilead. BJAR declares the receipt of research grants from Gilead and MSD and honoraria for advisory boards from AstraZeneca, Roche, Gilead, and F2G. KB has received research and educational grants from ViiV and Gilead, as well as consulting fees for advisory boards from ViiV, Gilead, MSD, and AstraZeneca. CR has received research grants from ViiV, Gilead, ZonMW, AIDSfonds, Erasmus MC, and Health~Holland, and honoraria for advisory boards from Gilead and ViiV. AR has received grants from the Bill and Melinda Gates Foundation and the Leids Universitair Fonds, participated on the board of an investor-initiated clinical trial on convalescent plasma for COVID-19, and is the chief editor of the Dutch Journal of Infectious Diseases and a member of the European Medicines Agency expert group on vaccines. All other authors declare no competing interests. 

5. In the online submission form, you indicated that data will be available upon request.

6. We note that you have indicated that there are restrictions to data sharing for this study. For studies involving human research participant data or other sensitive data, we encourage authors to share de-identified or anonymized data. However, when data cannot be publicly shared for ethical reasons, we allow authors to make their data sets available upon request. For information on unacceptable data access restrictions, please see http://journals.plos.org/plosone/s/data-availability#loc-unacceptable-data-access-restrictions. 

7. Please remove all personal information, ensure that the data shared are in accordance with participant consent, and re-upload a fully anonymized data set. 

Reviewers' comments:

Reviewer's Responses to Questions

**Comments to the Author**

1. Is the manuscript technically sound, and do the data support the conclusions?

Reviewer #1: Yes

Reviewer #2: Yes

Reviewer #3: Yes

2. Has the statistical analysis been performed appropriately and rigorously? 

Reviewer #1: Yes

Reviewer #2: Yes

Reviewer #3: Yes

3. Have the authors made all data underlying the findings in their manuscript fully available?

Reviewer #1: No

Reviewer #2: No

Reviewer #3: Yes

4. Is the manuscript presented in an intelligible fashion and written in standard English?

Reviewer #1: Yes

Reviewer #2: Yes

Reviewer #3: Yes

5. Review Comments to the Author

Reviewer #1: This study presents a prospective cohort study in adult people with HIV. The primary outcome was the SARS-CoV-2 spike-specific antibody level at 1, 6, 12, 18, and 24 months after completing a primary COVID-19 vaccination series. The authors compared the antibody kinetics over two years between PWH with a baseline CD4+ T-cell count <200 (n=16) vs. ≥200 (n=432) with a mixed-effects model. The authors conclude that long-term humoral responses were lower in PWH with a CD4+ T-cell count <200 compared to those with a CD4+ T-cell count ≥200.

This is a well-written manuscript presenting important data and emphasizing the importance of individual assessment of the need of COVID-19 vaccination among people living with HIV. See my comments below.

Abstract

Q1. Background: I would consider rephrasing the description of guideline recommendations given that both EACS, BHIVA and CDC are quite united in that priority should be giving those with advanced HIV-infection (low CD4 and detectable viral load). See EACS Guidelines 12.0 page 100, BHIVA (https://www.bhiva.org/SARS-CoV-2-vaccine-advice-for-adults-living-with-HIV-update) and CDC (https://www.cdc.gov/vaccines/covid-19/clinical-considerations/interim-considerations-us.html)

Q2. In the method section, lines 30-31, consider including a definition of primary series.

Introduction

Q3. Page 5, lines 51-52: Consider rephrasing. Some studies have shown higher mortality, some have not. Suggest adding references that represents both findings. Example: doi: 10.1097/QAD. 0000000000003129; doi: 10.1111/hiv.13174; doi: 10.1111/hiv.13515.

Q4. Page 5, line 63, first sentence, suggest adding references regarding earlier studies with follow-up.

Q5. Suggest adding a sentence in the Introduction that states current guidelines on vaccination in people with HIV (see examples above). In particular guidelines from the country the study was conducted in.

Method

Q6. Primary outcome is stated as The primary outcome was the level of S1-specific antibodies in PWH at 1, 6, 12, 18, and 24 months after primary vaccination.

Perhaps it should be clarified somehow “primary vaccination including/adjusted for the following boosters”? Because you do in fact account for the boosters?

Q7. In the subheading Clinical procedures, it says that data on comorbidities and co-medications were collected. I suggest presenting this in Table 1 and consider taken this into consideration in the statistical analysis or state why this was not deemed necessary.

Q8.Suggest stating more clearly the definition of Primary series vaccination in the Clinical procedures section where you state the different types of possible primary vaccinations. “Participants received two doses of BNT162b2, mRNA-1273, or ChAdOx1-S, or one dose of Ad26.COV2.S between January and August 2021 according to manufacturer’s regulations as part of the Dutch COVID-19 vaccination campaign”

Q9. In the Statistical analysis you state among variables added to the model “time since the last COVID-19 vaccination dose or last SARS-CoV-2 infection” .I guess this refers to the boosters? Perhaps it would be good to clarify that boosters were considered in the model.

Table 1:

Q10. History of a third SARSCoV-2 infection is marked with a b but it should perhaps be c?

Q11. In the first column, including all PWH, n=448. The sum in the column of “Covid-19 vaccines received” is just 403. Do you have missing data? Suggest updating table 1 so the numbers add up.

Q12. The same goes for History of at least one SARS-CoV-2 infection. The numbers do not add up. Missing data?

Q13. Same for: Dominant variant in the Netherlands at the time of first reported SARS-CoV-2 infection. Missing data?

Results

Q14. Page 15, lines 215-216: “Of these, one participant– a 61-year-old male – was admitted to the intensive care unit. His most recent CD4+ Tcell count was 316 cells per µL, and his nadir CD4+ T-cell count was 50 cells per µL.” This maybe reveals too much information about one patient and could potentially violate his integrity? This needs to be considered through the whole paragraph.

Discussion

Q15. The Discussion is quite short and could benefit from a more elaborate discussion of its findings and how it compares to other studies. For example, you state that this is the only study “this is the first longitudinal study in PWH extending beyond one year of follow-up after COVID-19 vaccination”. What is the follow-up time of the most recent studies?

Q16. Page 16, 222-225; here you conclude lower antibody levels after COVID-19 vaccination, as mentioned in Q9 it could be clearer what you mean by “COVID-19 vaccination”. Primary or primary including/adjusted for boosters?

Q17. You state in line 224 that the booster rates was comparable, but they are quite different in Table 1 between groups (primary + 2 boosters: 26, 7% vs 35,1%)?

Q18.Page 17, lines 249-251, please include references to these section

Limitations:

Q19. Suggest adding to the conclusion that the majority of your sample are men and how that effects generalizability.

Reviewer #2: Dear Dr. Roukens,

I appreciate the opportunity to review this interesting and important article longitudinal assessment of COVID-19 vaccine immunogenicity in people with HIV stratified by CD4+ T-cell count in the Netherlands: a two-year follow-up study. I enjoyed reading the manuscript. I commend the work on several strengths including:

1. Addressing the immunogenicity in PWH with CD4 < 200 and showing that this group needs to be prioritized. This is especially important during an epidemic and in low resource setting where prioritization is important.

2. Large sample size drawn from multiple centers, along with comprehensive follow-up

3. Longitudinal and prospective data collection, for an impressive duration of more than a year.

Considering the strengths of the manuscript, I noted a few areas where additional clarity would enhance its overall impact. Specifically, the paper could be further strengthened by incorporating more detailed information on the following points:

1. If the author could clarify the power calculations to clarify the need to include more participants, and the rationale for the inclusion of participants with a CD4 cell count of <500, along with precautions taken to minimize any potential bias with retrospective inclusion of participants. Furthermore, the inclusion of retrospective participants was not reflected in the figure. We would appreciate more clarification on the inclusion of the participants in the attached figure as well.

2. In the baseline results, there is a mention of a questionnaire, it is the first mention of a it being used in the study and was not mentioned in the methods. Please include it in the methods section as well with details as to how it was administered.

3. In the limitations, if you could also include the retrospective inclusion of participants could influence the results and precautions that were taken.

4. Additionally, if the authors could comment in manuscript, in the group with pre vaccination CD4 of >200, the participants who had a nadir of CD4 < 200 vs. the participants who had a nadir of CD4>200, would benefit from a booster. If it is not possible to calculate, can it be added to the limitations or the discussion (as per author’s discretion).

5. If the authors could clarify what was the cutoff age used to classify “higher age”, as referred to in the manuscript.

Best,

Soahum

Reviewer #3: Overall, I find this article interesting and valuable. The proposed study addresses an important and novel research question. The writing is easy to understand, and the flow of the text is good; the topic is clinically relevant and has ethics committee approval; there are some spelling and grammar errors that need revision.

6. PLOS authors have the option to publish the peer review history of their article (what does this mean? ). If published, this will include your full peer review and any attached files.

**Do you want your identity to be public for this peer review?** For information about this choice, including consent withdrawal, please see our Privacy Policy .

Reviewer #1: **Yes: ** Christina Carlander

Reviewer #2: **Yes: ** Soahum Bagchi

Reviewer #3: No

---

## [Author Response · Author response to Decision Letter 1]

22 Jan 2025

Response to Reviewers

We have ensured that our manuscript complies with the style requirements, including those for file naming.

This trial was funded by the Dutch Organization for Health Research and Development (ZonMw) [grant number 10430072010008].

The following statement is correct: "The funders had no role in study design, data collection and analysis, decision to publish, or preparation of the manuscript. " We have added this Role of Funder statement in our cover letter.

RDdV is supported by the Health~Holland grant EMCLHS20017, co-funded by the PPP Allowance made available by the Health~Holland, Top Sector Life Sciences & Health, to stimulate public–private partnerships, and is listed as an inventor of the fusion inhibitory lipopeptide [SARSHRC-PEG4]2-chol in a provisional patent application. KSH has received support for attending meetings and travel from Gilead. BJAR declares the receipt of research grants from Gilead and MSD and honoraria for advisory boards from AstraZeneca, Roche, Gilead, and F2G. KB has received research and educational grants from ViiV and Gilead, as well as consulting fees for advisory boards from ViiV, Gilead, MSD, and AstraZeneca. CR has received research grants from ViiV, Gilead, ZonMW, AIDSfonds, Erasmus MC, and Health~Holland, and honoraria for advisory boards from Gilead and ViiV. AR has received grants from the Bill and Melinda Gates Foundation and the Leids Universitair Fonds, participated on the board of an investor-initiated clinical trial on convalescent plasma for COVID-19, and is the chief editor of the Dutch Journal of Infectious Diseases and a member of the European Medicines Agency expert group on vaccines. All other authors declare no competing interests.

Please confirm that this does not alter your adherence to all PLOS ONE policies on sharing data and materials, by including the following statement: ""This does not alter our adherence to ONE policies on sharing data and materials.” (as detailed online in our guide for authors http://journals.plos.org/plosone/s/competing-interests). If there are restrictions on sharing of data and/or materials, please state these. Please note that we cannot proceed with consideration of your article until this information has been declared.

The following statement "This does not alter our adherence to PLOS ONE policies on sharing data and materials.” has been added to the Competing Interests statement and we have rewritten the last sentence to "The other authors have declared that no competing interests exist.” Find the cover letter for the full updated Competing Interests statement.

The full ethics statement is included in the ‘Methods’ section, including the full name of the ethics committee and that all participants provided written informed consent.

5. In the online submission form, you indicated that data will be available upon request.

We chose to share data under restricted access, as we are advised to do by our datamanagers. The data are pseudonymised data from real patients and we want to protect these maximally, as participants singed the informed consent saying that they cannot be identified from the data. According to our datamanagers this cannot be assured for pseudonymised data, as the data include potentially identifying and sensitive patient information.

6. We note that you have indicated that there are restrictions to data sharing for this study. For studies involving human research participant data or other sensitive data, we encourage authors to share de-identified or anonymized data. However, when data cannot be publicly shared for ethical reasons, we allow authors to make their data sets available upon request. For information on unacceptable data access restrictions, please see http://journals.plos.org/plosone/s/data-availability#loc-unacceptable-data-access-restrictions.

a) If there are ethical or legal restrictions on sharing a de-identified data set, please explain them in detail (e.g., data contain potentially identifying or sensitive patient information, data are owned by a third-party organization, etc.) and who has imposed them (e.g. a Research Ethics Committee or Institutional Review Board, etc.). Please also provide contact information for a data access committee, ethics committee, or other institutional body to which data requests may be sent.

We have updated our Data Availability statement in the submission form.

7. Please remove all personal information, ensure that the data shared are in accordance with participant consent, and re-upload a fully anonymized data set.

See our response to comment 6.

The reference list has been checked and updated.

Reviewers' comments:

Reviewer's Responses to Questions

Comments to the Author

1. Is the manuscript technically sound, and do the data support the conclusions?

Reviewer #1: Yes

Reviewer #2: Yes

Reviewer #3: Yes

2. Has the statistical analysis been performed appropriately and rigorously?

Reviewer #1: Yes

Reviewer #2: Yes

Reviewer #3: Yes

3. Have the authors made all data underlying the findings in their manuscript fully available?

Reviewer #1: No

Reviewer #2: No

Reviewer #3: Yes

4. Is the manuscript presented in an intelligible fashion and written in standard English?

Reviewer #1: Yes

Reviewer #2: Yes

Reviewer #3: Yes

5. Review Comments to the Author

Reviewer #1: This study presents a prospective cohort study in adult people with HIV. The primary outcome was the SARS-CoV-2 spike-specific antibody level at 1, 6, 12, 18, and 24 months after completing a primary COVID-19 vaccination series. The authors compared the antibody kinetics over two years between PWH with a baseline CD4+ T-cell count <200 (n=16) vs. ≥200 (n=432) with a mixed-effects model. The authors conclude that long-term humoral responses were lower in PWH with a CD4+ T-cell count <200 compared to those with a CD4+ T-cell count ≥200.

This is a well-written manuscript presenting important data and emphasizing the importance of individual assessment of the need of COVID-19 vaccination among people living with HIV. See my comments below.

Abstract

Q1. Background: I would consider rephrasing the description of guideline recommendations given that both EACS, BHIVA and CDC are quite united in that priority should be giving those with advanced HIV-infection (low CD4 and detectable viral load). See EACS Guidelines 12.0 page 100, BHIVA (https://www.bhiva.org/SARS-CoV-2-vaccine-advice-for-adults-living-with-HIV-update) and CDC (https://www.cdc.gov/vaccines/covid-19/clinical-considerations/interim-considerations-us.html)

Thank you for this suggestion. We have adjusted the background to make clear that guidelines indeed all prioritise PWH with advanced HIV or those who are clinically vulnerable. The background now reads as follows:

Although guidelines for COVID-19 additional vaccination strategies generally prioritize people with advanced HIV infection, recommendations vary globally, with some countries recommending an annual vaccination for all people with HIV (PWH), while others restrict this to PWH with a CD4+ T-cell count <200 cells per µL.

Q2. In the method section, lines 30-31, consider including a definition of primary series.

We have added a description of the primary COVID-19 vaccination series used in our participants.

See line 94-97: Between January and October 2021 participants received the primary vaccination series in accordance with manufacturers’ regulations as part of the Dutch COVID-19 vaccination campaign. The primary vaccination series consisted of two doses of BNT162b2, mRNA-1273, or ChAdOx1-S, or one dose of Ad26.COV2.S.

Introduction

Q3. Page 5, lines 51-52: Consider rephrasing. Some studies have shown higher mortality, some have not. Suggest adding references that represents both findings. Example: doi: 10.1097/QAD. 0000000000003129; doi: 10.1111/hiv.13174; doi: 10.1111/hiv.13515.

Thank you for your suggestion. We referred to a systematic review that we believed represents the highest level of evidence on this topic. This review included 27 studies on COVID-19 mortality in people with HIV (PWH), and one of the suggested references is also included in the review. The review found a slightly higher mortality risk in PWH compared to immunocompetent individuals (RR 1.20, 95% CI 1.05-1.36), and we have included the relative risk (RR) and confidence interval in the introduction of our revised manuscript.

Regarding the suggested references:

• doi: 10.1097/QAD.0000000000003129: This study is included in the systematic review.

• doi: 10.1111/hiv.13174: This reference is not included in the systematic review, but earlier results from the same author, based on a smaller cohort, are included in the review.

• doi: 10.1111/hiv.13515: This study is not included in the systematic review.

Q4. Page 5, line 63, first sentence, suggest adding references regarding earlier studies with follow-up.

We have added three references of earlier COVID-19 vaccination studies in PWH with follow-up.

Q5. Suggest adding a sentence in the Introduction that states current guidelines on vaccination in people with HIV (see examples above). In particular guidelines from the country the study was conducted in.

We have added the following sentence to the last paragraph of the introduction to address current guidelines on COVID-19 vaccination for people with HIV (PWH):

The lack of long-term data is reflected in the inconsistency of national guidelines regarding COVID-19 booster vaccination strategies for PWH. While the Centers for Disease Control and Prevention in the US recommend a yearly booster vaccination for PWH with a CD4+ T-cell count <200 cells per µL, European guidelines generally recommend yearly vaccination for all PWH, irrespective of their CD4+ T-cell count.

Method

Q6. Primary outcome is stated as: The primary outcome was the level of S1-specific antibodies in PWH at 1, 6, 12, 18, and 24 months after primary vaccination.

Perhaps it should be clarified somehow “primary vaccination including/adjusted for the following boosters”? Because you do in fact account for the boosters?

For the primary outcome, we did not adjust for booster vaccinations. However, as a secondary endpoint, we accounted for different variables in the model, including time since the last COVID-19 vaccination dose or last SARS-CoV-2 infection.

Q7. In the subheading Clinical procedures, it says that data on comorbidities and co-medications were collected. I suggest presenting this in Table 1 and consider taken this into consideration in the statistical analysis or state why this was not deemed necessary.

Thank you for addressing this as this was not fully correct. Comorbidities were not systematically collecte

---

## [Decision Letter · Decision Letter 1]

15 Apr 2025

Longitudinal assessment of COVID-19 vaccine immunogenicity in people with HIV stratified by CD4+ T-cell count in the Netherlands: a two-year follow-up study

PONE-D-24-47904R1

Dear Dr. Roukens,

We’re pleased to inform you that your manuscript has been judged scientifically suitable for publication and will be formally accepted for publication once it meets all outstanding technical requirements.

Kind regards,

Jianhong Zhou

Staff Editor

PLOS ONE

Additional Editor Comments (optional):

Reviewers' comments:

Reviewer's Responses to Questions

**Comments to the Author**

1. If the authors have adequately addressed your comments raised in a previous round of review and you feel that this manuscript is now acceptable for publication, you may indicate that here to bypass the “Comments to the Author” section, enter your conflict of interest statement in the “Confidential to Editor” section, and submit your "Accept" recommendation.

Reviewer #1: All comments have been addressed

Reviewer #2: All comments have been addressed

2. Is the manuscript technically sound, and do the data support the conclusions?

Reviewer #1: Yes

Reviewer #2: Yes

3. Has the statistical analysis been performed appropriately and rigorously? 

Reviewer #1: Yes

Reviewer #2: Yes

4. Have the authors made all data underlying the findings in their manuscript fully available?

Reviewer #1: No

Reviewer #2: No

5. Is the manuscript presented in an intelligible fashion and written in standard English?

Reviewer #1: Yes

Reviewer #2: Yes

6. Review Comments to the Author

Reviewer #1: (No Response)

Reviewer #2: Thank you for adressing the comments and edits.

The paper looks good, I appreciate the opportunity of being a part of this important publication.

Best regards,

7. PLOS authors have the option to publish the peer review history of their article (what does this mean? ). If published, this will include your full peer review and any attached files.

**Do you want your identity to be public for this peer review?** For information about this choice, including consent withdrawal, please see our Privacy Policy .

Reviewer #1: **Yes: ** Christina Carlander

Reviewer #2: **Yes: ** Soahum Bagchi

---

## [Editor Report · Acceptance letter]

PONE-D-24-47904R1

PLOS ONE

Dear Dr. Roukens,

I'm pleased to inform you that your manuscript has been deemed suitable for publication in PLOS ONE. Congratulations! Your manuscript is now being handed over to our production team.

Kind regards,

on behalf of

Dr. Jianhong Zhou

Staff Editor

PLOS ONE